# The kinase Isr1 negatively regulates hexosamine biosynthesis in *S. cerevisiae*

**Emma B. Alme**[1,2], **Erica Stevenson**[3,4,5], **Nevan J. Krogan**[3,4,5], **Danielle L. Swaney**[3,4,5], **David P. Toczyski**[1,2] *

**1** Department of Biochemistry and Biophysics, University of California San Francisco, San Francisco, California, United States of America, **2** Helen Diller Family Comprehensive Cancer Center, University of California San Francisco, San Francisco, California, United States of America, **3** Department of Cellular and Molecular Pharmacology, University of California San Francisco, San Francisco, California, United States of America, **4** California Institute for Quantitative Biosciences, University of California San Francisco, San Francisco, California, United States of America, **5** J. David Gladstone Institutes, San Francisco, California, United States of America

* dpt4darwin@gmail.com

**Data Availability Statement:** All relevant data are within the manuscript and its Supporting Information files. The mass spectrometry proteomics data have been deposited to the ProteomeXchange Consortium via the PRIDE

## Abstract

The *S. cerevisiae ISR1* gene encodes a putative kinase with no ascribed function. Here, we show that Isr1 acts as a negative regulator of the highly-conserved hexosamine biosynthesis pathway (HBP), which converts glucose into uridine diphosphate N-acetylglucosamine (UDP-GlcNAc), the carbohydrate precursor to protein glycosylation, GPI-anchor formation, and chitin biosynthesis. Overexpression of *ISR1* is lethal and, at lower levels, causes sensitivity to tunicamycin and resistance to calcofluor white, implying impaired protein glycosylation and reduced chitin deposition. Gfa1 is the first enzyme in the HBP and is conserved from bacteria and yeast to humans. The lethality caused by *ISR1* overexpression is rescued by co-overexpression of *GFA1* or exogenous glucosamine, which bypasses *GFA1's* essential function. Gfa1 is phosphorylated in an Isr1-dependent fashion and mutation of Isr1-dependent sites ameliorates the lethality associated with *ISR1* overexpression. Isr1 contains a phosphodegron that is phosphorylated by Pho85 and subsequently ubiquitinated by the SCF-Cdc4 complex, largely confining Isr1 protein levels to the time of bud emergence. Mutation of this phosphodegron stabilizes Isr1 and recapitulates the overexpression phenotypes. As Pho85 is a cell cycle and nutrient responsive kinase, this tight regulation of Isr1 may serve to dynamically regulate flux through the HBP and modulate how the cell's energy resources are converted into structural carbohydrates in response to changing cellular needs.

## Author summary

Protein phosphorylation is an essential regulatory mechanism that controls most cellular processes, integrating a variety of environmental signals to drive cellular growth. Yeast encode over 100 kinases, yet many remain poorly characterized. The *S. cerevisiae* gene *ISR1* encodes a putative kinase with no ascribed function. Here, we show that Isr1 decreases the synthesis of a critical structural carbohydrate, uridine diphosphate N-

partner repository with the dataset identifier PXD018429.

**Funding:** This work was supported by National Institutes of Health grants R35GM118104 to DPT and R01GM133981 to DLS. The funders had no role in study design, data collection and analysis, decision to publish, or preparation of the manuscript.

**Competing interests:** The authors have declared that no competing interests exist.

acetylglucosamine (UDP-GlcNAc), by mediating inhibition of one of the enzymes responsible for its synthesis, Gfa1. UDP-GlcNAc is the precursor to protein glycosylation, GPI anchor formation, and chitin synthesis, the first two of which are essential and conserved in humans. Throughout the cell cycle, and in response to changing environmental conditions, the cell must balance its use of glucose for energy production and generation of these structural carbohydrates. Here we show that Isr1 is regulated by both cell cycle and nutrient changes, and is rapidly degraded in a phosphorylation dependent manner. Isr1-mediated inhibition of UDP-GlcNAc synthesis may serve as a mechanism of dynamically regulating how the cell utilizes glucose in response to its environment.

## Introduction

Protein phosphorylation is a major signaling mechanism that is critical to the control of most cellular processes [1–3]. It is estimated that 75% of the proteome is phosphorylated and kinases are among the largest protein families, comprising about 2% of most eukaryotic genomes [4]. Kinases often act in cascades to amplify signals and there is significant cross-talk amongst protein kinases, allowing for the coordination of many cellular inputs. Despite their abundance and critical regulatory roles in the cell, many of the approximately 130 yeast kinases remain poorly characterized [5].

One such kinase is Inhibitor of Staurosporine Resistance 1 (Isr1). *ISR1* was first identified as a gene whose deletion is synthetically lethal with a temperature-sensitive allele of *PKC1* and whose overexpression sensitized cells to staurosporine, a broad spectrum kinase inhibitor that targets Pkc1 [6]. While the molecular basis for these phenotypes was unknown, they suggested a defect in cell wall homeostasis. Pkc1 is essential to the cell wall integrity pathway, which is critical to remodeling the cell wall in response to cellular growth and environmental stress [7]. Other studies identified *ISR1* as a gene whose overexpression causes sensitivity to caffeine, a general cell wall stress agent, and resistance to zymocin [8]. As chitin is the cellular receptor for zymocin [9], these phenotypes are consistent with reduced levels of chitin, a minor but critical structural component of the cell wall, comprising approximately 2% of the cell wall by mass [10]. The major protein component of the cell wall is glycosylphosphatidylinositol (GPI) anchors. Both GPI anchors and chitin share the same carbohydrate precursor, uridine diphosphate N-acetylglucosamine (UDP-GlcNAc), which is also the precursor for protein glycosylation [10]. Many proteins involved in cell wall carbohydrate biosynthesis are heavily glycosylated, making regulated production of this carbohydrate essential for proper cell wall homeostasis. N-glycosylation is also critical to many other cellular processes, including proper protein folding and trafficking as well as cell cycle progression [10,11]. The cellular requirement for UDP-GlcNAc changes throughout the cell cycle. While protein glycosylation occurs continuously as a cell grows, chitin synthesis predominates in G1 at the time of bud emergence and chitin deposition also greatly increases in response to cell wall stress [12–16].

UDP-GlcNAc is synthesized via the hexosamine biosynthesis pathway (HBP), which is highly conserved from bacteria to humans [17]. Most of the glucose imported into the cell is shunted into glycolysis after it is converted to fructose-6-phosphate, but approximately 6% enters the hexosamine biosynthesis pathway where it is converted into UDP-GlcNAc though the sequential action of four enzymes: glutamine: fructose-6-phosphate aminotransferase (Gfa1), GlcN-6-P acetyltransferase (Gna1), Phosphoacetyl-glucosamine mutase (Pcm1) and UDP-GlcNAc pyrophosphorylase (Qri1) [18]. The first step of this pathway, conversion of fructose-6-phosphate and glutamine to glucosamine-6-phosphate, is mediated by Gfa1 and is

the rate limiting step of the pathway [13,17]. Gfa1 activity has been shown to be directly proportional to UDP-GlcNAc synthesis and is regulated by nutrient and stress-responsive kinases [19–23].

We have identified a role for Isr1 as an inhibitor of the HBP, establishing a function for this kinase. High levels of Isr1 promote Gfa1 phosphorylation, and inhibition of the HBP by Isr1 is relieved by mutation of these Gfa1 phosphosites or bypassing Gfa1 by exogenous addition of glucosamine. Additionally, we have identified a critical Pho85-regulated SCF$^{CDC4}$ phosphodegron in Isr1. Mutation of this phosphodegron strongly affects HBP activity, highlighting how strict control of this kinase by post-translational modification can allow for signal integration and tight regulation of the HBP in response to changing cellular stimuli.

## Results

### Overexpression of Isr1 kinase activity is lethal

To begin to understand the role Isr1 might play in the cell, we first sought to test the effect of overexpressing *ISR1*. We placed the *ISR1* ORF under the control of the *GAL1* promoter on a 2μ plasmid. At this level of expression (referred to as GAL-2μ), Isr1 is lethal to the cell (Fig 1A). Isr1 is annotated as a putative protein kinase due to an unusual catalytic motif, HGD, in place of the signature HRD motif in its kinase domain. Mutation of the predicted aspartic acid proton acceptor in this motif to alanine abolished the lethality of GAL-2μ *ISR1*, indicating that kinase activity is required for Isr1 function (Fig 1A). Isr1 is a very low abundant protein [24] and difficult to overexpress for reasons that are currently unknown. The expression level of Isr1 from a 2μ plasmid utilizing the *ISR1* promoter (referred to as 2μ *ISR1*) is low compared to GAL-2μ *ISR1*, and even GAL-2μ *ISR1* does not result in extremely high levels of Isr1 (Fig 1B). To confirm this, we empirically determined the levels of Isr1 achieved by the GAL- 2μ *ISR1* construct. We epitope tagged Pkc1 expressed under its endogenous promoter using a 3x-Flag tag, the same epitope tag used for Isr1, and compared these proteins by western blot. This comparison demonstrates that GAL-2μ Isr1 protein levels are similar to the endogenous expression level of Pkc1 (Fig 1C).

### *ISR1* does not function in RNA processing

In contrast to the lethality we saw with GAL-2μ *ISR1*, in large scale studies using the *MATa* deletion collection, deletion of *ISR1* has resulted in few phenotypes [25]. *ISR1* deletion was shown to result in sensitivity to cordycepin, an adenosine analogue that causes premature termination, indicating a role in RNA processing [26]. However, we found that while the *isr1*Δ:: *KanMX* strain isolated from the *MATa* deletion collection did show sensitivity to cordycepin (Fig 2A), this was not rescued by integrating *ISR1* at the endogenous locus in this strain. By contrast, we observed that *isr1*Δ was resistant to tunicamycin, an N-glycosylation inhibitor [27]. This phenotype was not previously reported and sensitivity to tunicamycin was restored by integrating *ISR1* in the *isr1*Δ strain (Fig 2A). This indicates that tunicamycin resistance, but not cordycepin sensitivity, is dependent on Isr1 activity. The end of the *ISR1* coding sequence overlaps with the 3'UTR of its neighboring gene, *YTH1*, which is an essential component of the mRNA cleavage and polyadenylation factor. Thus, the cordycepin sensitivity of the *isr1*Δ strain from the deletion collection is likely a result of a reduction in function of *YTH1*.

Importantly, many of the genetic interactions reported for *ISR1* from large scale studies using the *MATa* deletion collection are likely due to this neighboring gene effect as well, making it difficult to discern which phenotypes are specific to *ISR1*. This caveat of large scale collections has been previously described and the neighboring gene effect is estimated to result in incorrect annotation of approximately 10% of genes [28]. We generated our own *isr1*Δ strain

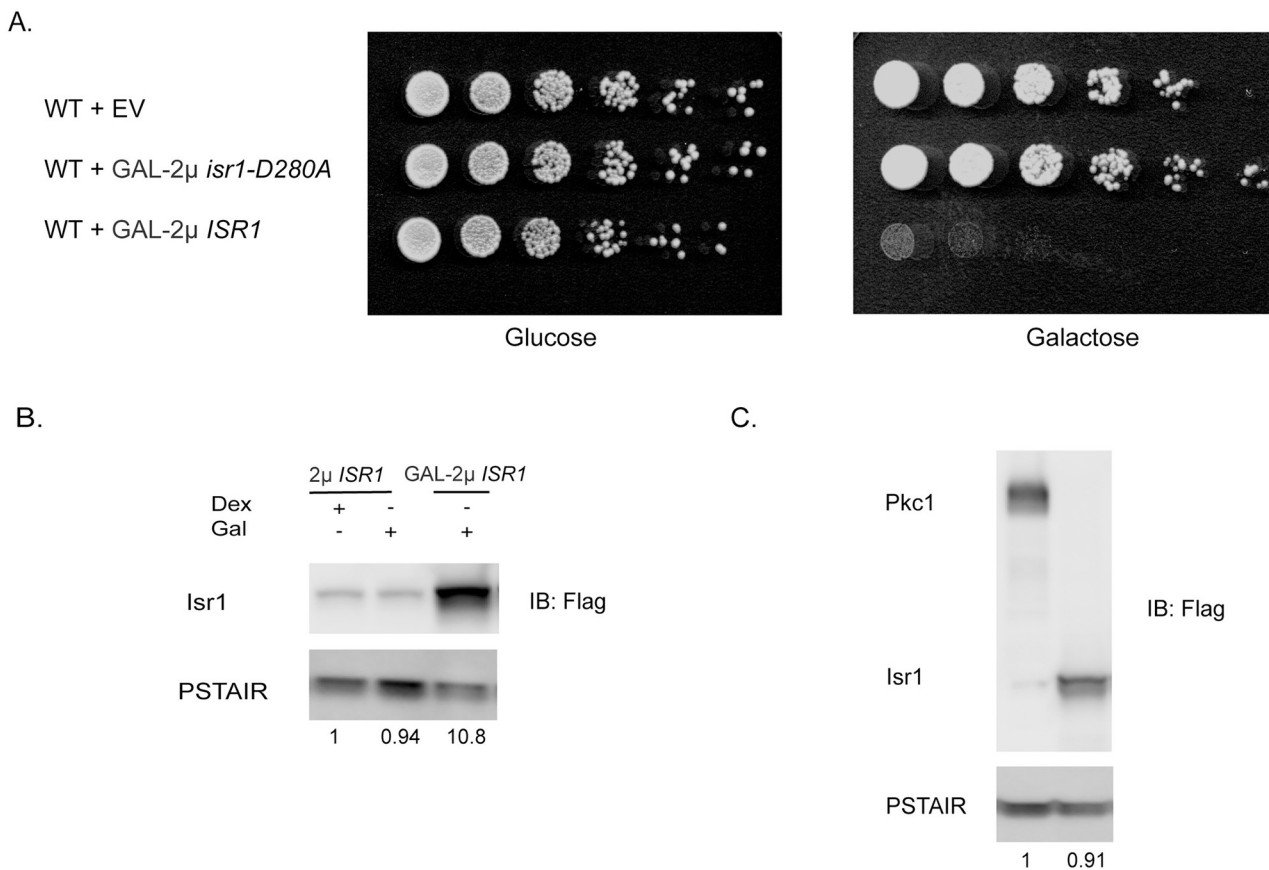

**Fig 1. Overexpression of *ISR1* is lethal.** (A) Wild type cells were transformed with an empty vector (EV) or 2µ plasmid expressing either a wild-type (GAL-2µ *ISR1*) or a kinase-dead allele of *ISR1* (GAL-2µ *isr1-D280A*) from the *GAL1* promoter. Cells were serially diluted on YPD or YPGal plates. (B) Western blot showing relative expression level of Isr1-3xFlag expressed on a 2µ plasmid from either its endogenous promotor (2µ *ISR1*) or the *GAL1* promoter (GAL-2µ *ISR1*). Values represent the normalized relative band intensity for Isr1. PSTAIR is a monoclonal antibody that recognizes the PSTAIR sequence in Cdc28 and is shown as a loading control. Cells were inoculated into YM-1 with the indicated carbon source and grown for 4 hours under NAT selection. **(C)** Relative levels of Isr1-3xFlag expressed from the *GAL1* promoter compared to Pkc1-3xFlag tagged at the endogenous locus.

using a hygromycin selectable marker and found that this strain did not exhibit sensitivity to cordycepin, but did show resistance to tunicamycin (Fig 2A). We speculate that the HygMX sequence used here might function as a more stable 3'UTR for *YTH1* than the KanMX sequence used to generate the *isr1Δ* strain in the deletion collection. This *isr1Δ::HygMX* strain is used in all subsequent experiments.

### *ISR1* overexpression has phenotypes associated with a deficiency in UDP-GlcNAc

Given the resistance to tunicamycin that we uncovered with our newly generated *isr1Δ* strain, we wished to determine if higher levels of *ISR1*, reciprocally, caused tunicamycin sensitivity. Because of the lethality associated with GAL-2µ *ISR1*, we performed this characterization utilizing 2µ *ISR1*. We added a NAT selectable marker to the plasmid to allow drug screening on rich media containing Nourseothricin (NAT) instead of synthetic dropout media, allowing for lower concentrations of drugs to be used. 2µ overexpression of *ISR1*, but not a kinase dead allele, *isr1-D280A*, resulted in sensitivity to tunicamycin (Fig 2B). Tunicamycin acts as an ER-stress agent by inhibiting the first step in N-glycosylation, which involves the transfer of N-

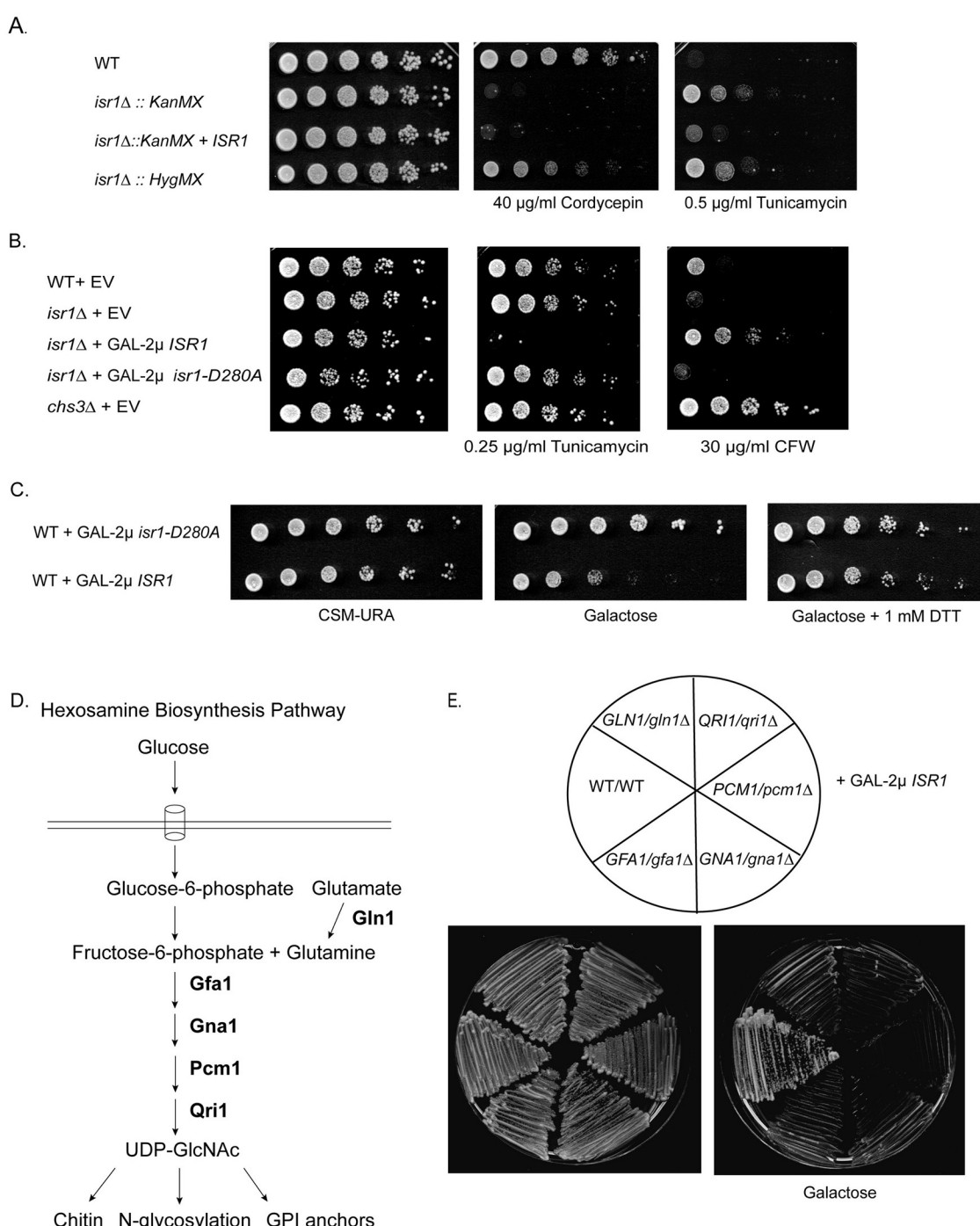

**Fig 2. Altering *ISR1* dosage confers phenotypes associated with changes in flux through the hexosamine biosynthesis pathway.**
(A) *isr1Δ::KanMX*, isolated from the *Mat a* deletion collection, exhibits a sensitivity to cordycepin that is not shared by *isr1Δ::HygMX* generated in this study. Both exhibit resistance to tunicamycin. Strains of the indicated genotypes were grown on YPD alone or containing 40 μg/ml cordycepin or 0.5 μg/ml tunicamycin. (B) 2μ *ISR1* causes sensitivity to tunicamycin and resistance to calcofluor white (CFW). Strains of the indicated genotypes were transformed with EV, or a 2μ plasmid expressing either a wild-type (2μ *ISR1*) or a kinase-dead allele of *ISR1* (2μ *isr1-D280A)* from the *ISR1* promoter. Strains were spotted on CSM in the presence or absence of 0.25 μg/ml tunicamycin or 30 μg/ml CFW. (C) Overexpression of *ISR1* does not cause sensitivity to DTT. Wild-type cells were transformed with GAL-2μ *ISR1* or GAL-2μ *isr1-D280A* and spotted as in B on CSM-URA plates containing glucose, galactose or galactose + 1 mM DTT. (D) The hexosamine biosynthesis pathway converts fructose-6-phosphate and glutamine into UDP-GlcNAc, the precursor to N-glycosylation, GPI-anchors, and chitin. Gln1 acts upstream of the HBP and is required for glutamine synthesis. (E) GAL-2μ *ISR1* is synthetically lethal with all the enzymes in the HBP. Diploid heterozygous deletions of the

enzymes in the HBP were generated and transformed with GAL-2μ *ISR1* and struck on YPD or YPGal plates. Note that wild-type diploids are less sensitive than haploid strains to *ISR1* overexpression.

acetylglucosamine-1-phosphate (GlcNAc-1-P) from UDP-N-acetylglucosamine (UDP-Glc-NAc) to the carrier lipid dolichyl-phosphate (Dol-P) [27,29]. Thus, the sensitivity to tunicamycin in response to *ISR1* overexpression could be due to a defect in N-glycosylation or a general increase in ER-stress. To distinguish between these possibilities, we tested if *ISR1* overexpression conferred sensitivity to other ER stress agents and found that GAL-2μ *ISR1* did not confer sensitivity to DTT, suggesting that the tunicamycin sensitivity is not due to a general increase in ER-stress (Fig 2C). For reasons that are currently unknown, the GAL-2μ *ISR1* phenotype is less severe on synthetic media, allowing this sensitivity test to be performed.

We also found that 2μ *ISR1* conferred resistance to calcofluor white (Fig 2B), recapitulating previous data indicating a role for *ISR1* in cell wall integrity [8]. Calcofluor white is a cell-wall stress agent that targets chitin and resistance indicates a specific deficiency in chitin deposition in the cell wall. Deletion of the main chitin synthase, *CHS3*, results in complete resistance to calcofluor white, whereas other cell wall mutants or overproduction of chitin results in sensitivity [9,30–32].

Both chitin and protein glycosylation are produced from the same precursor carbohydrate, UDP-GlcNAc, which is also the precursor for GPI anchors (Fig 2D). Chitin and GPI anchors predominantly function at the cell surface: GPI anchors are the major protein component of the cell wall, whereas chitin is a more minor, but still critical structural component [10,14]. N-glycosylation functions on the cell surface as well, but is also essential for proper protein folding and trafficking [11]. Previous studies have shown that blocking GPI anchor formation increases chitin deposition in the cell wall [33]. UDP-GlcNAc synthesis is the rate-limiting step in the production of chitin, which is a polymer of UDP-GlcNAc, and GPI-anchor inhibition is thought to allow the accumulation of excess UDP-GlcNAc that is no longer converted into GPI anchors [13,33].

The tunicamycin sensitivity of the 2μ *ISR1* strain and the tunicamycin resistance observed in the *isr1Δ* strain might be due to misregulation of protein N-glycosylation. If *ISR1* overexpression specifically inhibited N-glycosylation, one would expect a larger pool of available UDP-GlcNAc for chitin synthesis, resulting in a compensatory increase in chitin production. However, we observed that 2μ *ISR1* caused resistance to calcofluor white, indicating a deficiency in chitin deposition. This is consistent with a defect in both protein glycosylation and chitin biosynthesis. This suggest that Isr1 acts upstream of N-glycosylation, reducing UDP-GlcNAc production.

## *ISR1* overexpression is synthetically lethal with enzymes in the hexosamine biosynthesis pathway

Most of the glucose imported into the cell is shunted into glycolysis after it is converted to fructose-6-phosphate, but approximately 6% enters the hexosamine biosynthesis pathway (HBP), where fructose-6-phosphate and glutamine are converted into UDP-GlcNAc though the sequential action of four enzymes: Gfa1, Gna1, Pcm1, and Qri1 (Fig 2D) [18]. This pathway is highly conserved from bacteria to humans [17]. Given that each of these enzymes are essential, we generated heterozygote deletions in genes encoding each of these enzymes and expressed GAL1-*ISR1* in these strains to test the genetic interactions of this pathway with *ISR1*. We found that GAL-2μ *ISR1* is synthetically lethal with all four enzymes in the pathway (Fig 2E). Notably, diploid wild-type cells are less sensitive to GAL-2μ *ISR1* as compared to haploid cells.

Gln1 is not part of the HBP, but synthesizes glutamine, one of the precursors to the pathway [34], and also exhibits synthetic sickness in combination with GAL-2μ *ISR1*. These genetic interactions suggest that *ISR1* overexpression inhibits the HBP.

## *ISR1* lethality is rescued by exogenous glucosamine, but not other precursors to UDP-GlcNAc synthesis

The first enzyme of this pathway, Gfa1, converts fructose-6-phosphate and glutamine to glucosamine-6-phosphate (Fig 3A) [35]. This is the rate-limiting step of the pathway and Gfa1 activity is proportional to chitin synthesis [13]. *GFA1* essentiality can be bypassed by exogenous glucosamine, which can be phosphorylated by hexokinases. Thus, *gfa1*Δ strains are glucosamine auxotrophs. Addition of exogenous glucosamine drives flux through the HBP and increases UDP-GlcNAc production [35,36]. We tested if the lethality associated with GAL-2μ *ISR1* overexpression can be rescued by addition of exogenous glucosamine and found that in the presence of 5mM glucosamine, GAL-2μ *ISR1* no longer inhibited growth (Fig 3B). Some cell wall mutants have growth phenotypes that can be rescued by both exogenous glucosamine and by addition of sorbitol to the media [36]. As sorbitol is an osmotic stabilizer, this implies that the slow growth phenotype of such mutants is caused by cell lysis due to a weakened cell wall. To determine if a similar general cell wall defect is responsible for the lethality associated with GAL-2μ *ISR1*, we tested growth of this strain in the presence of 10% sorbitol and found that sorbitol was not capable of rescuing growth (S1 Fig). This suggests that the lethality of GAL-2μ *ISR1* is due to a growth arrest caused by a deficiency of UDP-GlcNAc production and not cell lysis caused by a weakened cell wall.

As expected, addition of glucosamine also restored sensitivity of the 2μ *ISR1* overexpression strain to calcofluor white by driving flux through the HBP and increasing the amount of UDP-GlcNAc (Fig 3C). By contrast, glucosamine did not restore sensitivity to *chs3*Δ (Fig 3C), as an increase in UDP-GlcNAc cannot restore chitin synthesis in the absence of the chitin synthase itself. Thus, the resistance to calcofluor white observed upon 2μ overexpression of *ISR1* is unlikely to be due to a general trafficking defect that mislocalizes Chs3. This is in agreement with the lack of sensitivity observed upon exposure to DTT. Similarly, exogenous glucosamine restored growth to the 2μ *ISR1* strain in the presence of tunicamycin (Fig 3C). The ability of glucosamine to rescue *ISR1* overexpression phenotypes implies that Isr1 acts at or before the first step of the HBP. To determine if a deficiency in one of the precursors to the HBP is responsible for the *ISR1* overexpression phenotypes, we tested if fructose and glutamine, the reactants in the production of glucosamine-6-phosphate (Fig 3A), could rescue the *ISR1* overexpression phenotypes. We also tested alternative or non-fermentable carbon sources, which bypass glycolysis entirely, and found that neither the precursors to the HBP nor non-fermentable carbon sources were capable of rescuing *ISR1* overexpression phenotypes (Fig 3C and 3D, S1 Fig). The ability of glucosamine, but not precursors to the HBP, to completely rescue *ISR1* overexpression phenotypes indicates that Isr1 likely inhibits the first step of the HBP.

## Isr1 is a negative regulator of Gfa1

Our observation that exogenous glucosamine rescues the lethality associated with GAL-2μ *ISR1* overexpression suggests that Isr1 might inhibit flux through the HBP. If this is the case, overexpression of *GFA1* should rescue this lethality as well. To explore this possibility, we co-overexpressed *GFA1* on the same plasmid as *GAL1-ISR1* and as expected, *GFA1* overexpression also rescued the lethality associated with GAL-2μ *ISR1* overexpression (Fig 4A). To determine if this was unique to *GFA1*, we co-overexpressed another enzyme downstream in this

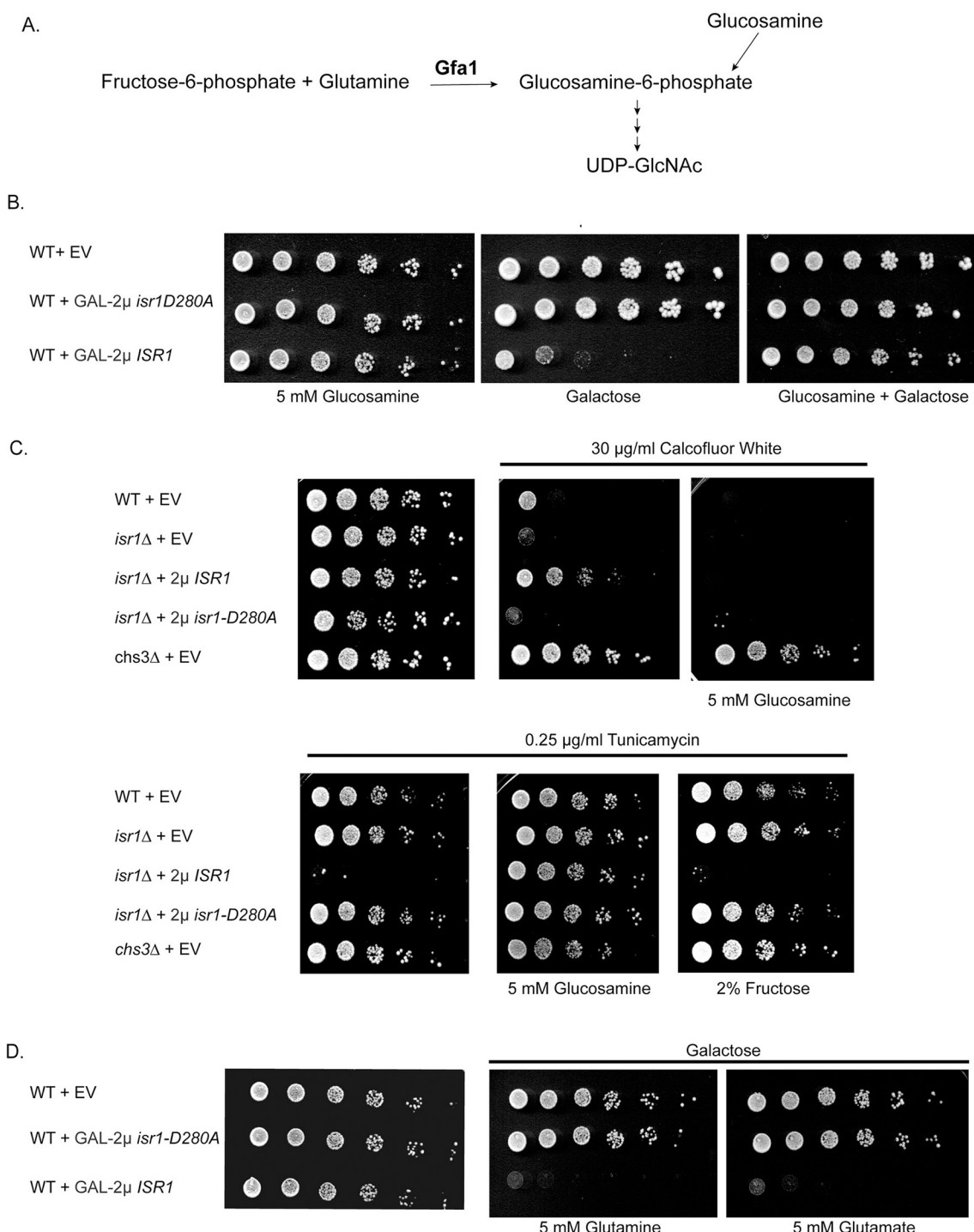

**Fig 3. Exogenous glucosamine specifically rescues *ISR1* overexpression phenotypes.** (A) Gfa1, the first and rate limiting enzyme in the HBP, converts fructose-6-phosphate and glutamine into glucosamine-6-phosphate. *GFA1* essentiality can be bypassed by addition of exogenous glucosamine. (B) Exogenous glucosamine rescues GAL-2μ *ISR1* lethality. Wild-type cells were transformed with GAL-2μ *ISR1* or GAL-2μ *isr1-D280A* and spotted onto CSM + 5mM glucosamine, CSM galactose, or CSM galactose + 5mM glucosamine. (C) Exogenous glucosamine, but not fructose, restores resistance to tunicamycin and sensitivity to CFW. Strains of the indicated genotypes were transformed with EV, 2μ ISR1, or 2μ *isr1-D280A*. Strains were spotted on CSM alone or containing 0.25 μg/ml tunicamycin, 30 μg/ml CFW, 5 mM glucosamine, or 2% fructose as indicated. (D) Lethality of GAL-2μ *ISR1* is not rescued by precursors to the HBP. Experiment was performed as in B and strains were spotted on CSM alone or CSM + galactose containing 5 mM glutamine or glutamate (MSG).

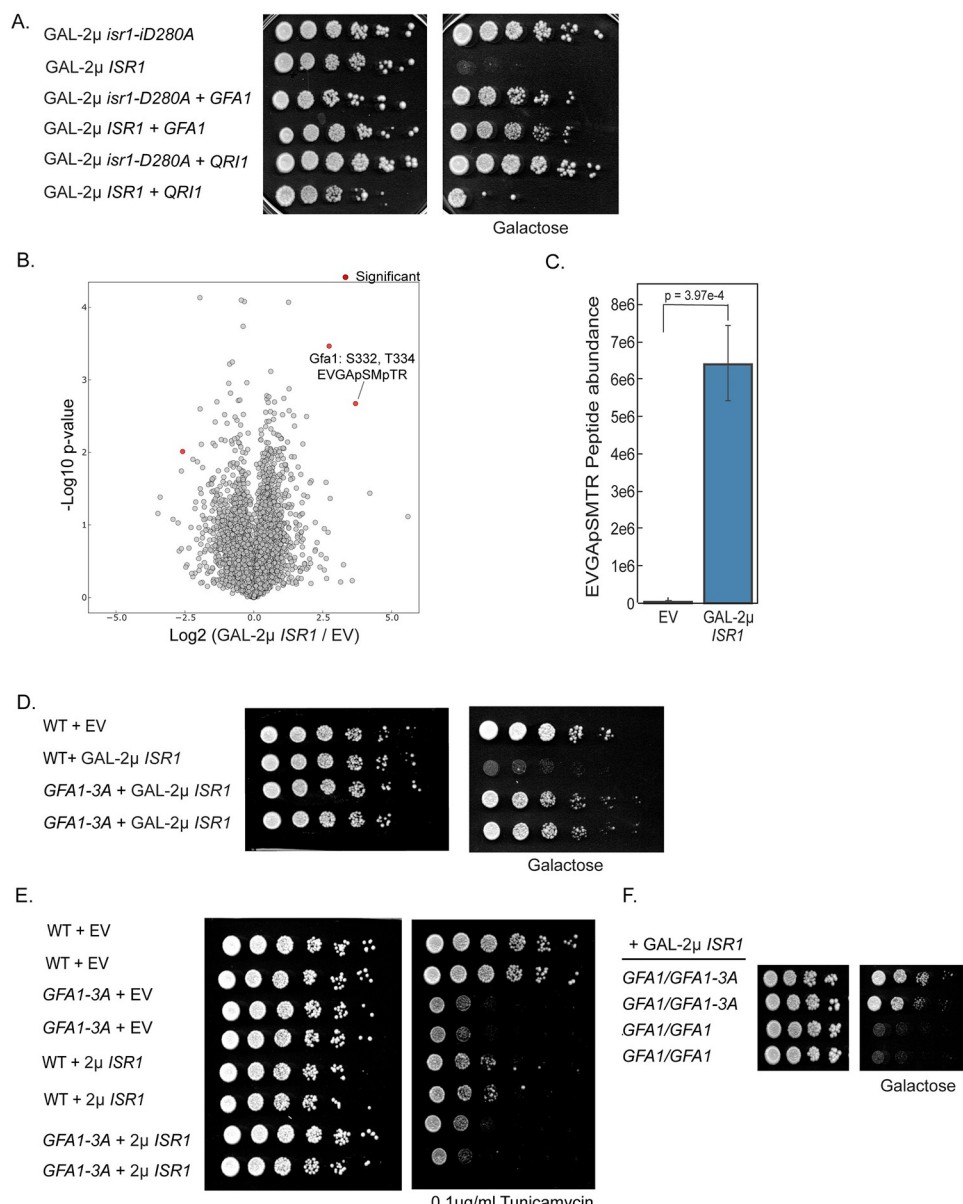

**Fig 4. Gfa1 is phosphorylated in an Isr1-dependent manner and a *GFA1* phosphomutant rescues *ISR1* lethality.**
(A) *GFA1* overexpression rescues the lethality of GAL-2μ *ISR1*. Wild-type cells were transformed with a 2μ plasmid expressing either *ISR1* or *isr1-D280A* from the GAL1 promoter and either *GFA1* or *QRI1* from their endogenous promoters. Cells were serial diluted onto YPD or YPGal. (B) The Gfa1 S332 S334 phosphopeptide is enriched in cells expressing GAL-2μ *ISR1*. Volcano plot of phosphopeptides detected in *isr1Δ* cells expressing EV or GAL-2μ *ISR1*. Peptides with a p < 0.01 and having a fold change +/- 4-fold different between are considered significant. (C) Gfa1 S332 is phosphorylated in an Isr1-dependent manner. Targeted data extraction for Gfa1 S332 phosphorylation sites, which was only detected in the Gal-2u *ISR1* conditions. P-value is the result of a two-sided unpaired t-test. (D) Mutation of *GFA1* phosphorylation sites at the endogenous locus ameliorates the lethality of GAL-2μ *ISR1*. Wild-type cells or a *GFA1* phosphomutant (S332A,T334A, S336A) were transformed with EV or GAL-2μ *ISR1* and spotted as in A. (E) The *GFA1-3A* allele is hypomorphic. Cells of the indicated genotypes were transformed with EV or 2μ *ISR1* and serial diluted on YPD in the presence or absence of 0.1 μg/ml tunicamycin. (F) Experiment was performed as in D with diploid cells of the indicated genotypes.

pathway, Qri1, with *GAL1-ISR1*. *QRI1* only mildly increased growth on galactose, suggesting that Isr1 acts at the level of Gfa1 within the HBP (Fig 4A).

In other organisms, including *C. albicans*, drosophila, and humans, phosphorylation by multiple kinases, including AMPK and PKA, has been shown to both positively and negatively regulate Gfa1 activity [19,20,22,23,37,38]. Overexpression of *ISR1* did not alter Gfa1 protein levels (S2 Fig), suggesting that it might act by inhibiting Gfa1 function. However, the *GFA1-TAP* allele was unable to support viability and therefore it remains possible that the epitope tagged *GFA1* allele is stabilized or cannot be targeted by Isr1.

A previous mass spectrometry study of kinase interactors identified Gfa1 as a physical interactor with Isr1 in three separate immunoprecipitation experiments [39]. Given this, we sought to determine if Gfa1 is phosphorylated in an Isr1-dependent manner. We conducted phosphoproteomics in *isr1Δ* cells expressing either an empty vector or GAL-2μ *ISR1* and detected over 8,779 distinct phosphorylated peptides on 1,860 proteins (S1 Table). In this experiment, we detected 4 sites of phosphorylation on Gfa1: S199, S253, S332, and T334. Of these sites, the doubly phosphorylated peptide containing S332 and T334 was 12.9-fold up-regulated in *ISR1*-expressing cells (Fig 4B). Additionally, a singly phosphorylated S332 peptide was also observed only in the *ISR1*-expressing cells. Targeted data extraction for this peptide revealed its phosphorylation level was 179-fold higher in *ISR1*-expressing cells (Fig 4C). To examine the importance of these phosphorylations, we mutated S332, T334, and S336 to alanine. S336 was included because it was adjacent to the other two sites and was seen to be phosphorylated in a previous whole-phosphoproteome screen [40]. Mutation of S332, T334, and S336 to alanine at the endogenous locus (referred to as *GFA1-3A*) rescued the lethality associated with GAL-2μ *ISR1* overexpression (Fig 4D). This suggests that Isr1 inhibits Gfa1 by promoting its phosphorylation.

Notably, the *GFA1-3A* mutant was slightly hypomorphic. While the *GFA1-3A* strain had no growth defect under normal conditions, it was sensitive to tunicamycin and resistant to calcofluor white (Fig 4E, S2C Fig). This sensitivity was recessive, as expected for a hypomorph, and was epistatic to 2μ *ISR1* (Fig 4D, S2 Fig). By contrast, the *GFA1*-3A mutant was also dominant in its ability to rescue GAL-2μ *ISR1*-induced lethality: the *GFA1/GFA1-3A* heterozygous diploid was resistant to GAL-2μ *ISR1* (Fig 4F). Our finding that the resistance conferred by *GFA1-3A* is dominant is consistent with the hypothesis that *GFA1-3A* rescues *ISR1* overexpression because it is refractory to inhibition by Isr1. Thus, while GFA1-3A may have slightly less activity than wild type in the uninhibited/unphosphorylated state, this still likely represents significantly more activity than the phosphorylated state. These data suggest Isr1 is a negative regulator of Gfa1.

It should be noted that, while the *GFA1* phosphomutant rescued GAL-2μ *ISR1*-induced lethality, *GFA1-3A* cells overexpressing *ISR1* still grew more slowly than *GFA1-3A* cells overexpressing *isr1-D280A*. Therefore, while our data support a model where Isr1 inhibits the HBP by promoting the phosphorylation of Gfa1, Isr1 may have additional substrates, additional sites on Gfa1, or both.

### Isr1 is an unstable cell-cycle regulated protein targeted by Pho85 and Cdc4

Given that even mild overexpression of Isr1 inhibits the essential function of Gfa1, we next sought to understand how Isr1 is regulated in the cell. The cellular need for UDP-GlcNAc varies throughout the cell cycle and we therefore tested if Isr1 is a cell cycle regulated protein. Analysis of Isr1 protein levels after arresting cells in each phase of the cell cycle showed that Isr1 protein levels are largely confined to the G1/S transition and are drastically reduced in G1 and M phases (Fig 5A). This matches known transcriptional data for *ISR1* [41]. Previous work

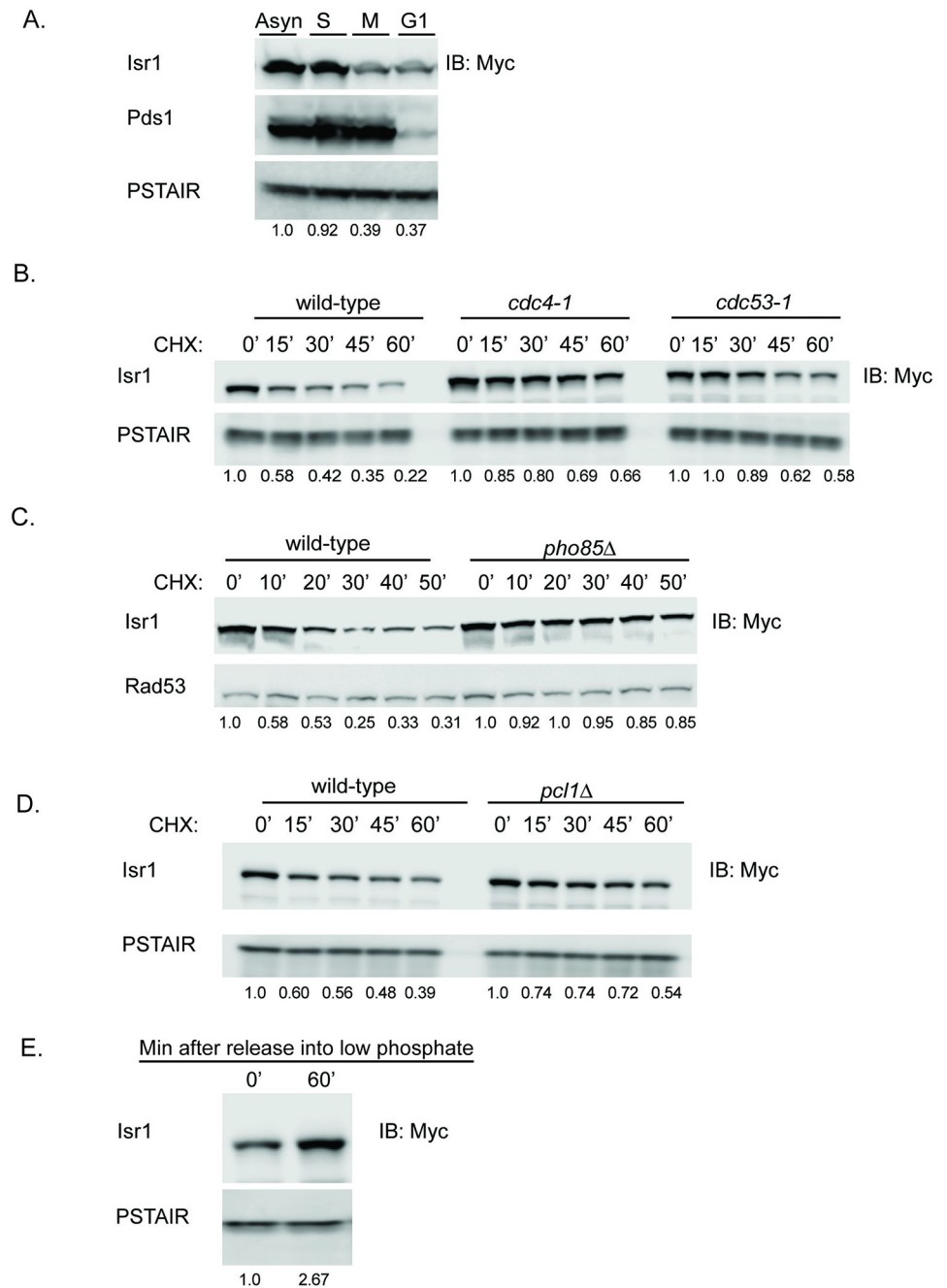

**Fig 5. Isr1 is an unstable protein targeted by Pho85 and Cdc4.** (A) Isr1 is cell cycle regulated. Western blot showing levels of Isr1-13xmyc in cells growing asynchronously or arrested in G1, S, or M phase with alpha factor, Hydroxyurea (HU) or nocodazole, respectively. PSTAIR and Pds1 are shown as loading and cell cycle arrest controls, respectively. PSTAIR is a monoclonal antibody that recognizes the PSTAIR sequence in Cdc28. In all panels, values represent the normalized relative band intensity for Isr1. (B) Isr1 is targeted for degradation by Cdc4. Cycloheximide-chase assay of Isr1-13xmyc in wild-type, *cdc4-1*, or *cdc53-1* strains. Cells were shifted to the non-permissive temperature for 30 minutes before addition of cycloheximide for the indicated number of minutes. Levels of Isr1-13xMyc and PSTAIR (loading control) are shown. (C) Degradation of Isr1 is dependent on Pho85. Experiment was performed as in B at 30°C in *PHO85* or *pho85Δ* cells. Rad53 is shown as a loading control (D) Isr1 is stabilized by deletion of *PCL1*. Experiment as in B at 30°C in *PCL1* or *pcl1Δ* cells. (E) Western blot showing levels of Isr1-13xMyc and PSTAIR (loading control) after shifting cells to low phosphate media for 60 minutes.

from our lab suggested that Isr1 is a substrate of Cdc4 [42]. Cdc4 is an F box protein that functions as a substrate adaptor for the SCF ubiquitin ligase complex and targets many substrates for proteasome-mediate degradation in a cell-cycle dependent manner [43]. Consistent with this, a cycloheximide chase of Isr1-13xMyc in wild-type cells, as compared to temperature-sensitive mutants of *CDC4* and *CDC53*, showed that Isr1 is unstable, with a half-life of approximately 30 minutes, and is stabilized by inactivation of *CDC4* or *CDC53* (Fig 5B).

Cdc4 recognizes and binds a phosphodegron with the optimal sequence (S/T)-P-X2-4 –(S/T) in which both (S/T) are phosphorylated, often by CDK kinases [44–47]. Isr1 was previously identified as an in vitro substrate of Pho85 [48], a CDK-like kinase that can function with any of 10 different cyclins in response to changes in nutrient and cell-cycle conditions [49,50]. Isr1 was shown to be phosphorylated by Pho85 in complex with Pcl1, which functions in G1 progression [48,51]. We found that deletion of *PHO85* stabilized Isr1, suggesting that Pho85-phosphorylation of Isr1 enables Isr1 to be targeted by Cdc4 for degradation (Fig 5C). Deletion of *PCL1* also partially stabilized Isr1 (Fig 5D). Given this partial stabilization, it remains possible that other Pcls, in addition to Pcl1, can also function with Pho85 to target Isr1. Pho85 is inhibited by the CDK inhibitor Pho81 in low phosphate conditions [52]. Upon release into low phosphate media, Isr1 levels increased, as would be expected if Pho85 activity promotes its instability (Fig 5E). Pho85 activity has also been shown to regulate carbohydrate metabolism [49,53]. We therefore tested if Isr1 stability was altered by carbon source. We found that growth in glycerol, a non-fermentable carbon source, as well as glucose withdrawal, also partially stabilized Isr1, whereas growth in galactose did not (S3 Fig). This supports a model where Pho85-mediated degradation of Isr1 might allow Isr1 activity to be responsive to environmental conditions.

## Stabilization of an endogenous Isr1 phosphodegron recapitulates Isr1 overexpression phenotypes

Examination of the Isr1 protein sequence revealed two putative Cdc4 phosphodegrons in the N-terminus of the protein, each comprising two CDK consensus phosphorylation sites separated by several amino acids (Fig 6A). We constructed an Isr1 phosphodegron mutant *(ISR1-PD)* by mutating these four residues (T4, S8, T86, S92) as well as two adjacent serine/threonine residues to alanine at the endogenous locus. The *ISR1-PD* mutant was completely stable (Fig 6B), strongly supporting a model where phosphorylation of Isr1 at these sites targets it for degradation by Cdc4.

We next tested if this regulation is functionally relevant by examining the calcofluor white sensitivity of the *ISR1-PD* mutant. We found that these mutations are sufficient to cause resistance to calcofluor white, consistent with there being higher levels of Isr1 activity in *ISR1-PD* mutants (Fig 6C). This same resistance was recapitulated on Congo Red, a cell-wall stress agent that also targets chitin (S4 Fig) [54]. Similarly, the *ISR1-PD* mutant was also highly sensitive to tunicamycin (Fig 6D). Taken together, these data show that the endogenous phosphodegron mutant of Isr1 recapitulates *ISR1* overexpression phenotypes, supporting a model where tight regulation of Isr1 is critical for controlling its function in the cell. Interestingly, deletion of the entire N-terminal 93 amino acids of Isr1, which includes both phosphodegrons, was also sensitive to tunicamycin (Fig 6D). This implies that this truncation mutant is active and the only function of the N-terminus of Isr1 is in mediating its instability. Our mass spectrometry data also identified an additional proline-directed phosphorylation site in Isr1, S47, that may also be targeted by Pho85 or could represent autophosphorylation.

To determine the physiological relevance of Isr1 regulation, we tested the genetic interactions of this Isr1 phosphodegron mutant with deletions of the enzymes of the HBP pathway.

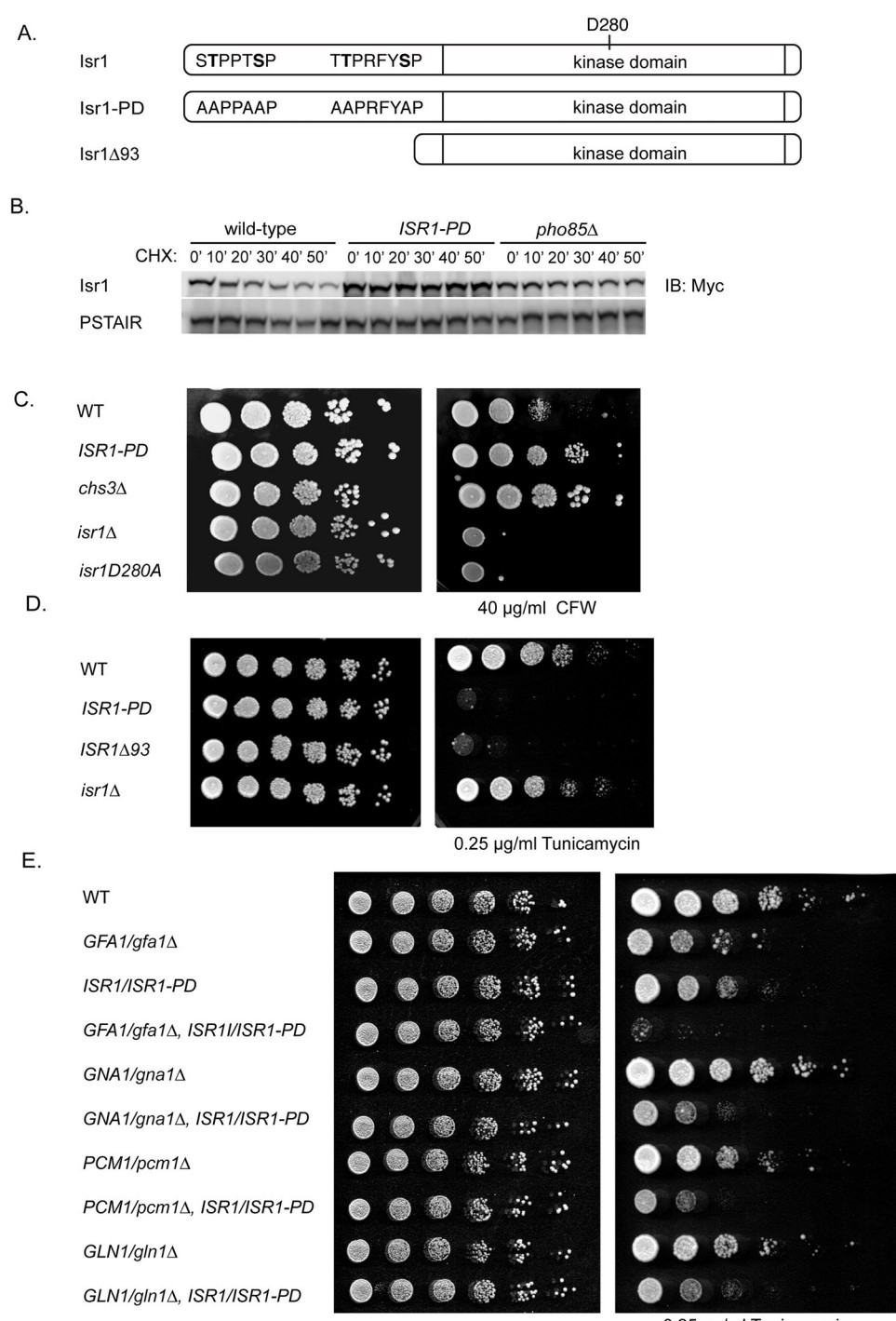

**Fig 6. Stabilization of an endogenous Isr1 phosphodegron recapitulates overexpression phenotypes.** (A) Diagram of Pho85 and Cdc4 consensus sites in Isr1 that comprise a phosphodegron. Six sites (T3, T4, S8, T85, T86, S92) are mutated to alanine at the endogenous locus in the *ISR1-PD* mutant. (B) An Isr1 phosphodegron mutant is stable. Cycloheximide-chase assay of Isr1-13xmyc in wild-type, *pho85Δ*, or cells expressing *ISR1-PD*. Cycloheximide was added for the indicated number of minutes. Levels of Isr1-13xMyc and PSTAIR (loading control) are shown. PSTAIR is a monoclonal antibody that recognizes the PSTAIR sequence in Cdc28. (C) An Isr1 phosphodegron mutant is resistant to calcofluor white (CFW). Strains of the indicated genotypes were diluted onto YPD with or without 40 μg/ml CFW. (D) An Isr1 phosphodegron mutant is sensitive to tunicamycin. Experiment was performed as in C, except strains were spotted on YPD with or without 0.25 μg/ml tunicamycin. (E) Diploid strains of the indicated genotypes were diluted onto YPD with or without 0.25 μg/ml tunicamycin.

Because these genes are essential, we examined diploids that were heterozygous for each HBP gene, as well as the *ISR1-PD* mutant. We found that *ISR1/ISR1-PD* showed a level of sensitivity to tunicamycin similar to that seen for *GFA1/gfa1Δ* (Fig 6E). One interpretation of this is that one copy of the stabilized Isr1 phosphodegron mutant reduces Gfa1 activity by half. Notably, The *GFA1/gfa1Δ* strain grew more slowly on tunicamycin than deletions of the other three enzymes in the pathway, as would be expected given that Gfa1 is the rate-limiting step of the pathway. While *ISR1/ISR1-PD* and *GFA1/gfa1Δ* were virtually synthetically lethal in the presence of tunicamycin, this synthetic sickness was not observed when *ISR1-PD* was combined with other deletions in the pathway. The specificity of this synthetic interaction further supports a model where Isr1 activity negatively regulates Gfa1 function and emphasizes the large effect of an even minor increase in Isr1 stabilization on flux through the HBP.

## Discussion

Throughout the cell cycle and in response to changing nutrient conditions, the cell must balance its utilization of glucose for energy needs and as a precursor to structural and signaling carbohydrates. Here, we uncover a function for a poorly characterized kinase, Isr1, as a negative regulator of the hexosamine biosynthesis pathway (HBP) that synthesizes UDP-GlcNAc from glucose.

Only ~15% of yeast genes are phenotypic when overexpressed [55]. The lethality of overexpressing *ISR1*, combined with the limited phenotypes of its deletion, support a model where Isr1 is a negative regulator of an essential protein. Our data firmly establishes a role for Isr1 in inhibition of the HBP based on several lines of evidence. First, 2µ overexpression of *ISR1* results in resistance to calcofluor white and sensitivity to tunicamycin, whereas *isr1Δ* is slightly sensitive to calcofluor white and resistant to tunicamycin. These opposing phenotypes are consistent with a deficiency in UDP-GlcNAc when *ISR1* is overexpressed and an increase in UDP-GlcNAc when *ISR1* is absent. Genes that function in protein trafficking or the downstream utilization of UDP-GlcNAc in protein glycosylation are expected to have a compensatory increase in chitin production and to have a sensitivity to other ER-stress agents [33]. This was not observed for *ISR1* overexpression, suggesting that it inhibits the HBP itself. Consistent with this, we found that *ISR1* overexpression is synthetically lethal with every enzyme in the HBP.

All phenotypes of *ISR1* overexpression are rescued by either overexpressing *GFA1*, the enzyme of the first step of the HBP, or by bypassing *GFA1* function by the addition of exogenous glucosamine. Addition of reactants upstream of Gfa1, such as fructose and glutamine, had no effect. This strongly suggested that Isr1 inhibits this first step in the HBP. Furthermore, we found that phosphorylation of a specific serine in Gfa1 is dependent on Isr1 activity and mutation of this phosphorylation site to alanine rendered cells insensitive to *ISR1* overexpression. Thus, Isr1 appears to act as a negative regulator of Gfa1. The *GFA1* phosphomutant is hypomorphic, yet epistatic to *ISR1* overexpression and dominant in its ability to rescue GAL-2µ *ISR1* lethality, strongly implying that Isr1 exerts its function in the cell by promoting Gfa1 phosphorylation.

Isr1 may phosphorylate Gfa1 directly, as suggested by the existing immunoprecipitation-mass spectrometry data, or it might act in a kinase cascade. However, our phosphoproteomics dataset did not reveal specific activation of other kinases that might act downstream of Isr1 and deletion of several stress-responsive kinases did not rescue *ISR1* lethality (S5 Fig). Notably, deletion of the majority of the N-terminal domain of Isr1 rendered Isr1 more active, implying that this portion of the protein is not required for substrate specificity. Given that the

remaining residues comprise little more than a kinase domain, it remains unclear how Isr1 targets its substrates.

ISR1 was previously implicated in cell wall homeostasis by Mehlgarten et al.'s observation that *ISR1* overexpression results in zymocin resistance, a cell wall stress agent [8]. The findings of that paper suggested a role for Isr1 in chitin biology and are in agreement with the function we have uncovered for Isr1 as a negative regulator of Gfa1. Mehlgarten et al. delete *PHO85* and find this abolishes the *ISR1* overexpression phenotype, leading the authors to posit that Pho85 activates Isr1. However, our results are directly contrary to that hypothesis and we suggest that the rescue that Mehlgarten et al. observed is due to the pleiotropic effects that Δpho85 has on the cell wall.

Additional genetic evidence suggests an interaction between *ISR1* and *GFA1*: Isr1 was first identified due to the fact that its overexpression resulted in heightened sensitivity to staurosporine, an inhibitor of Pkc1, and *ISR1* deletion is synthetically lethal with a temperature sensitive allele of *PKC1* [6]. Pkc1 is known to positively regulate *GFA1* transcription through Rlm1 activation [21,56]. Therefore, the synthetic lethality of *ISR1* and *PKC1* is consistent with a role for Isr1 as a negative regulator of Gfa1. Similarly, *ISR1* has been reported in multiple studies to have strong negative genetic interaction with *PRR1* [3,57,58], a kinase that is a physical interactor with Gfa1 [39]. These data would be consistent with Prr1 and Isr1 independently regulating Gfa1. Notably, *prr1Δ* does not rescue GAL-2μ *ISR1* lethality (S5 Fig).

The human homologue of Gfa1, Gfat1, has been previously shown to be negatively regulated by phosphorylation via AMPK and PKA [19,23,37]. Thus, inhibition of Gfa1 via phosphorylation may be a convergent or conserved mechanism of regulating hexosamine biosynthesis in response to changes in nutrients. The cellular requirement for UDP-GlcNAc changes throughout the cell cycle. While protein glycosylation occurs continuously as a cell grows, chitin synthesis predominates in G1 at the time of bud emergence [12]. Indeed, Gfa1 expression increases during G1 or in response to pheromone or cell wall stress [21,33,41]. Thus, dynamic control of Gfa1 is required to properly shunt glucose into or away from the hexosamine biosynthesis pathway as needed. Notably, Isr1 protein levels peak at the G1/S transition, just after the period of time during the cell cycle at which higher levels of chitin synthesis are required. Given that Gfa1 activity is directly proportional to chitin synthesis [13], Isr1 might provide a mechanism of rapidly returning the cell to a normal level of chitin synthesis following bud emergence. Gfa1 is a highly abundant protein, whereas Isr1 is expressed at very low levels. We speculate that tight regulation of Isr1 protein levels via Pho85 and Cdc4 might allow for rapid dynamic changes in Gfa1 activity and could provide the cell with a less energetically costly mechanism of regulating Gfa1 function than degrading Gfa1 itself. In this way, Isr1 could function to integrate environmental signals to regulate the remodeling of the cell wall and glycoprotein biosynthesis. While more studies are required to determine the specific environmental inputs that modulate Isr1 instability and the mechanism by which Isr1 inhibits Gfa1, this work firmly establish a cellular function for Isr1 as a key negative regulator of an essential, conserved pathway.

## Materials and methods

### Strains and plasmids

Strains and plasmids used in this study are listed in S2 and S3 Tables, respectively. All yeast strains in this study are in the S288C background. Unless otherwise noted, single gene deletions are from the *Mat a* deletion collection (Open Biosystems) and TAP-tagged strains were a gift from Erin O'Shea and Jonathan Weismann. Cloning of constructs and transformations were done using standard techniques. Diploid heterozygous deletions were made by

transformation of a *KanMX* cassette with homology to the promoter and 3' UTR of the gene of interest into a wild type diploid or heterozygous *ISR1-PD* diploid strain. The *ISR1-PD* and *GFA1-3A* strains were constructed by gene replacement: point mutants were synthesized on a plasmid containing a nourseothricin (NAT) (ISR1) or *HygMX (GFA1)* selection marker (S2 Table, S6 Fig). A PCR fragment was generated with homology to the endogenous promoter at the 5' end and the MX cassette at the 3' end. This PCR fragment was transformed into the deletion strain to replace the gene deletion previously marked by *HygMX (ISR1)* or *KanMX (GFA1)*. Integrations were screened by marker loss/gain and then verified by PCR. In the case of *GFA1*, *GFA1-3A* was first generated in a *GFA1/gfa1Δ* diploid and haploid mutants were isolated by tetrad analysis.

## Yeast cell culture

Selective media lacking specific amino acids or nucleobases was made using Complete Supplement Mixture (CSM) from Sunrise Sciences, supplemented with yeast nitrogen base and 2% dextrose. Unless otherwise noted, cells were cultured at 30˚C on YM-1 media supplemented with 2% dextrose or CSM-URA to maintain plasmid selection. 100 µg/ml NAT was added to maintain plasmids when strains were grown in YM-1. Expression from the *GAL1* promoter was induced with 2% (wt/vol) galactose for 4 hours. For experiments with temperature-sensitive strains, cells were maintained at 23 ˚C and shifted to 37˚C for 30 minutes before initiating the experiment. Low phosphate media was made as with CSM, but using YNB lacking amino acids and phosphate (Formedium CYN0803) and supplemented with 0.55g KCl/liter. To measure Isr1 protein levels in low phosphate, cells were washed several times in low-phosphate media and resuspended in low-phosphate media for 60 minutes.

## Western blotting

From cultures in midlog phase, cell pellets of equivalent optical densities were collected, washed with 1 mL 4˚C $H_2O$, and frozen on dry ice. Standard TCA precipitations were preformed to extract proteins. Samples were resuspended in SDS-PAGE sample buffer, boiled for 5 min, and cell lysates were analyzed by SDS-PAGE using 4–20% gradient Tis-HCl gels (BioRad, #3450034). Proteins were transferred onto 0.2-µm nitrocellulose membrane. Western blots were performed with low-salt PBS with Tween-20 (PBS-T) (15 mM NaCl, 1.3 mM $NaH_2PO_4$, 5.4 mM $Na_2HPO_4$, 0.05% Tween-20). Primary antibody incubations were performed in 5% (wt/vol) nonfat dry milk and low-salt PBS-T. Antibodies were used as follows: α-Rad53 (Abcam ab104232); α-Flag (Sigma-Aldrich, F3165); α-Myc (BioLegend, #626802), α-PSTAIR (Sigma-Aldrich P7962), α-Pds1 (generous gifts from Adam Rudner), α-TAP (Thermo-Scientific CAB1001). Western blots were visualized by the LiCor Odyssey Imaging System. Isr1 levels were quantified using the Image Studio Lite software and intensity values were normalized to the loading control for each lane.

## Cell cycle arrest

Cells were grown overnight in YM-1 and inoculated to OD600 = 0.2 and grown at 30˚C for 45 minutes. For arrest in G1, 15 µg/ml alpha factor was added. After 90 minutes, an additional 10 µg/ml alpha factor was added. Cells were harvested after 2.5 hours. For arrest in S and M phase respectively, 200 mM HU or 15 µg/ml nocodazole was added and cells were harvested after 2 hours. Arrest was confirmed by microscopic analysis and equivalent ODs were processed for western blots as described above.

## Cycloheximide-chase assays

Cells were grown as for western blotting to midlog phase. Cycloheximide was added to cultures for a final concentration of 50 μg/ml after collection at the t = 0 time point. Equivalent ODs were collected for each time point and were processed for western blots as described above. For *cdc53-1* and *cdc4-1* experiments, cells (including wild-type control) were shifted to 37 ˚C for 30 minutes before addition of cycloheximide. For alternative carbon sources, cells were grown overnight in YM-1 supplemented with 2% dextrose, 2% galactose or 2% glycerol/1% ethanol, inoculated at 0.2 OD, and grown to mid-log phase before addition of cycloheximide.

## Mass spectrometry sample preparation

Cells were grown overnight in YM-1 supplemented with 2% raffinose and 100 μg/mL NAT and then inoculated at 0.3 OD. After 1 hour, 40% galactose was added to a final concentration of 2% galactose and cells were grown for an additional 3 hours to OD ~0.8. 40 OD of cells were washed 2x with water and flash frozen in liquid nitrogen. Cells pellets were combined and lysed in a denaturing urea buffer (8 M urea, 0.1 M ammonium bicarbonate, pH 8, 150 mM NaCl, 1 Roche mini protease inhibitor tablet without EDTA/10 ml, ½ Roche phosSTOP tablet/ 10 ml) using $14 \times 1.5$ min bursts on a BioSpec mini bead-beater at room temperature. 2 ml screw-cap tubes used for lysis were pierced with an 18-gauge needle and spun in a swinging bucket centrifuge for 30 s at $1000 \times g$ to collect extract. Extract was rotated end-over-end for 30 min at room temperature before clarification via centrifugation at $17,000 \times g$ for 7 min followed by a second centrifugation at $17,000 \times g$ for 2 min, both at room temperature. Extracts were quantitated using a BCA protein quantification kit (Pierce). 1 M TCEP (Sigma C4706–2) was added to final concentration of 4 mM to 1 mg of protein and incubated for 30 min at room temperature. 0.5 M iodoacetamide (Sigma L1149–5G, prepared fresh in water) was added to a final concentration of 10 mM and incubated in the dark for 30 min. To quench excess iodoacetamide, 0.5 M DTT was added to a final concentration of 10 mM for another 30 min in the dark. Samples were diluted ∼fourfold (to less than 2 M urea) with 0.1 M Tris, pH 8, and Lys-C/ trypsin (Promega, Madison, WI, V5071, dissolved in 50 mM acetic acid) was added at a ratio of 200 μg trypsin to 1 mg total protein. Samples (∼1 ml total volume of diluted sample) were incubated for 20 hours at room temperature with rotation. After digestion, TFA was added to a final concentration of 0.3–0.1% TFA, with pH of final solution ~2.

## Phosphopeptide enrichment by immobilized metal affinity chromatography

Iron nitrilotriacetic acid (NTA) resin were prepared in-house by stripping metal ions from nickel nitrilotriacetic acid agarose resin (Qiagen) with 100 mM ethylenediaminetetraacetic (EDTA) acid, pH 8.0 three times. Resin was washed twice with water and 100 mM iron(III) chloride was applied three times. The iron-NTA resin was washed twice with water and once with 0.5% formic acid. Iron-NTA beads were resuspended in water to create a 25% resin slurry. 60 μl of Fe-NTA resin slurry was transferred to individual Silica C18 MicroSpin columns (The Nest Group) pre-equilibrated with 100 μl of 80% ACN, 0.1% TFA on a vacuum manifold. Subsequent steps were performed with the Fe-NTA resin loaded on the Silica C18 columns. Peptide samples were mixed twice with the Fe-NTA resin and allowed to incubate for 2 minutes. The resin was rinsed four times with 200 μl of 80% ACN, 0.1% TFA. In order to equilibrate the chromatography columns, 200 μl of 0.5% formic acid was applied twice to the resin and columns. Peptides were eluted from the resin onto the C18 column by application of 200 μl of 500 mM potassium phosphate, pH 7.0. Peptides were washed twice with 200 μl of

0.5% formic acid. The C18 columns were removed from the vacuum manifold and eluted twice by centrifugation at 1000*g* with 60 μl of 50% ACN, 0.25% TFA. Peptides were dried with a centrifugal adaptor and stored at -20˚C until analysis by liquid chromatograph and mass spectrometry.

## Proteomic data acquisition and analysis

Peptides were resuspended in 45 uL of 4% formic acid, 3% ACN, and 2uL was analyzed in on a Bruke timsTOF Pro mass spectrometry system equipped with a Bruker nanoElute high-pressure liquid chromatography system interfaced via a captiveSpray source. Samples were directly injected on a C18 reverse phase column (25 cm x 75 μm packed with ReprosilPur C18 AQ 1.9 um particles). Peptides were separated by an organic gradient from 2 to 28% ACN at 400nl/min over the course of a 120 acquisition. Each sample was injected twice, once with data-dependent PASEF acquisition to build a spectral library, and one via a diaPASEF acquisition for quantitative analysis. The mass spectrometry proteomics data have been deposited to the ProteomeXchange Consortium via the PRIDE [59] partner repository with the dataset identifier PXD018429. All data-dependent PASEF files [60] were search against the Saccharomyces cerevisiae proteome database (Downloaded from the Saccharomyces Genome Database January 13, 2015). Peptide and protein identification searches were performed with the Spectronaut Pulsar software (www.spectronaut.org) to generate a spectral library of detected phosphorylated peptides with a 1% false-discovery rate at the peptide and protein level. Spectronaut was further used with the default settings to analyzed the diaPASEF data [61] and extract quantitative regulation of detected phosphorylation sites. Label-free quantification and statistical testing of phosphorylation sites was performed using the artMS R-package (version 1.3.7) (https://github.com/biodavidjm/artMS), and the MSstats statistical R-package (version 3.16.0) [62]. Instant Clue [63] was used for figure generation of proteomics data.

## Spot tests

Yeast strains were inoculated into 3–5 ml YM-1 or CSM-URA + 2% dextrose grown overnight with aeration at 30˚. Fivefold dilution series were set up in 96-well plates, and 3–5 μl aliquots of the dilution series were transferred to YPD, YPGAL or CSM plates. Drug concentrations and alternative carbon sources are specified in individual figures. In experiments utilizing plasmids overexpressing *ISR1*, plates also contained 100 μg/ml NAT to maintain plasmid selection. Plates were incubated 2–3 days at 30˚ until colonies formed and then were photographed.

## Supporting information

**S1 Fig. *ISR1* lethality is not rescued by osmotic stabilizers or other carbon sources.** (A) Lethality of GAL-2μ *ISR1* is not rescued by sorbitol. Cells of the indicated genotypes expressing EV or GAL-2μ *ISR1* were diluted onto YPD or YPGal + 10% sorbitol. (B) Lethality from GAL-2μ *ISR1* overexpression is not rescued by alternative carbon sources. Wild-type cells were transformed with EV, GAL- 2μ *ISR1* or GAL- 2μ *isr1-D280A* and spotted onto CSM containing 2% glucose, galactose or the indicated carbon source. (C) Experiment performed as in B, but with 3% glycerol.
(TIF)

**S2 Fig. *ISR1* does not affect Gfa1 protein levels.** (A). 2μ *ISR1* does not affect Gfa1 protein levels at any cell cycle stage. Cells were inoculated in CSM-URA and arrested in G1, S, and M phase with alpha factor, HU or nocodazole respectively. Note that Gfa1 was upregulated in response to alpha factor, as expected. (B) GAL-2μ *ISR1* does not alter Gfa1 protein levels. Cells

were grown overnight in CSM-URA raffinose and inoculated in CSM-URA galactose for 4 hours. (C) Cells of the indicated genotypes were transformed with EV or 2μ *ISR1* and serial diluted on YPD with or without 30 μg/ml calcofluor white. (D) Diploids of the indicated genotypes were transformed with EV or 2μ *ISR1* and spotted on YPD with or without 0.25 μg/ml tunicamycin.
(TIF)

**S3 Fig. Isr1 is partially stabilized in non-fermentable carbon sources.** (A) Isr1 is partially stabilized by glycerol/ethanol, but not galactose. Cycloheximide-chase assay of Isr1-13xmyc grown in YM-1 containing 2% dextrose, 2% galactose or 2% glycerol/1% ethanol. Cells were grown overnight in the indicated carbon source, inoculated in fresh media, and cycloheximide was added for the indicated number of minutes. For the no glucose condition, cells were grown in YM-1 with dextrose, washed twice in media without a carbon source, and suspended in media with no carbon source at the same time as adding cycloheximide.
(TIF)

**S4 Fig. Mutation of an Isr1 phosphodegron confers resistance to Congo Red.** (A) An Isr1 phosphodegron mutant is resistant to Congo Red. Strains of the indicated genotypes were diluted onto YPD in the presence or absence of 150 μg/ml Congo Red.
(TIF)

**S5 Fig. MAPKs and *PRR1* are not downstream of *ISR1*.** (A) Strains of the indicated genotypes expressing Gal- 2μ *ISR1* or GAL-2μ *isr1-D280A* (wild-type only) were struck on YPD or YPGal plates. (B) *ppr1Δ* cells expressing EV or GALl-2μ *ISR1* were serial diluted onto YPD or YPGal.
(TIF)

**S6 Fig. *ISR1* phosphodegron mutant sequence.** Double-stranded DNA sequence used to generate *ISR1* phosphodegron mutant. Blue sequence is the end of the *ISR1* promoter. Lowercase red base pairs indicate mutated residues.
(DOCX)

**S1 Table. Phosphoproteomics dataset for *isr1Δ* vs GAL-2μ *ISR1*.** Phosphopeptide dataset comparing *isr1Δ* cells expressing either EV or GAL-2μ *ISR1*. The "Results" tab contains the full set of quantitative results displayed in the volcano plot (Fig 4B). The "S332" tab contains the quantitative data displayed in Fig 4C.
(XLSX)

**S2 Table. Strains used in this study.**
(DOCX)

**S3 Table. Plasmids used in this study.**
(DOCX)

## Acknowledgments

We thank Jonathan Asfaha and David Morgan for reagents and generously allowing use of their equipment. We'd also like to thank the members of the Toczyski lab for helpful discussions.

## Author Contributions

**Conceptualization:** Emma B. Alme, David P. Toczyski.

**Formal analysis:** Danielle L. Swaney.

**Funding acquisition:** Danielle L. Swaney, David P. Toczyski.

**Investigation:** Emma B. Alme, Erica Stevenson, Danielle L. Swaney.

**Methodology:** Emma B. Alme, Danielle L. Swaney, David P. Toczyski.

**Resources:** Nevan J. Krogan, Danielle L. Swaney, David P. Toczyski.

**Supervision:** Nevan J. Krogan, David P. Toczyski.

**Writing – original draft:** Emma B. Alme.

**Writing – review & editing:** Emma B. Alme, Danielle L. Swaney, David P. Toczyski.

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
