## [Decision Letter · Decision Letter 0]

7 Apr 2020

Dear Dr Toczyski,

Thank you very much for submitting your Research Article entitled 'The kinase Isr1 negatively regulates hexosamine biosynthesis in S. cerevisiae' to PLOS Genetics. Your manuscript was fully evaluated at the editorial level and by independent peer reviewers. The reviewers appreciated the attention to an important topic but identified some aspects of the manuscript that should be improved.

All three reviewers have had the opportunity to read and review your submission, and there is consensus that this is a strong paper that communicates important findings that will be of interest to those studying cell signaling pathways and the regulation of cellular metabolism. We also agree with reviewers that several minor issues need to be addressed prior to publication. However we believe this can be addressed with changes to the text of the manuscript, and will not require additional experimentation. Some of the issues raised that you should address in the revised manuscript include:

1. Reviewer 1 points out some inaccuracies in the literature, surrounding the evolutionary relationship between ISR1 and Raf. Considering the points made by Reviewer 1, the authors are advised to make significant changes to the introduction and discussion of the manuscript in order to address these concerns. (See Reviewer 1, point #1 and point #5.)

2. All three reviewers raise important questions about some of the immunoblot data presented in the paper. Specifically, Reviewer 1 points out that the use of PSTAIR as a loading control requires some clarification. Furthermore, Reviewers 2 and 3 point out that some of the blots shown (e.g., Figure 1B and Figure 5) should include quantitation.

3. Both Reviewer 2 and Reviewer 3 commented on the somewhat unexpected phenotype of the GFA1-3A mutant. Although the reviewers agree that the genetic tests shown in Figure 4F support the authors’ interpretations, it is still unclear why the GFA1-3A and the isr1 deletion exhibit similar phenotypes. Perhaps this point warrants additional discussion.

Reviewer 3 also points out that the data in this paper do not provide evidence that Isr1 directly phosphorylates Gfa1. While such evidence would strengthen the assertions of the manuscript, there is consensus that this should not be a criteria for publication. Additionally, there are several other comments made by reviewers that the authors should attempt to address with minor revisions to the text of the manuscript. 

We therefore ask you to modify the manuscript according to the review recommendations before we can consider your manuscript for acceptance. Your revisions should address the specific points made by each reviewer.

[LINK]

Yours sincerely,

Jason MacGurn

Guest Editor

PLOS Genetics

Gregory P. Copenhaver

Editor-in-Chief

PLOS Genetics

Reviewer's Responses to Questions

**Comments to the Authors:**

Reviewer #1: Manuscript: PGENETICS-D-20-00243

TITLE: The kinase Isr1 negatively regulates hexosamine biosynthesis in S. cerevisiae

AUTHOR(S): David P Toczyski

Summary. Yeast encode 117 protein kinases in the superfamily of eukaryotic protein kinases, plus a smattering of atypical protein kinases related in function but not evolutionary lineage. Almost all of the yeast kinases have been characterized at least to some extent. This paper determines the function of one of the few remaining poorly characterized protein kinases in yeast. The experiments presented here show the following 1) they demonstrate the role of ISR1 in regulating the hexosamine pathway; 2) they use phosphoproteomic data to identify GFA1 as the ISR1 substrate and 3) they show the regulation of ISR1 abundance via a PHO85-targeted phosphodegron. These results are significant and novel. A few changes to the manuscript that could strengthen the paper are noted below.

1. ISR1 and Raf/Ras signaling. The first paper to describe ISR1 is the Miyahara paper from 1998. This paper states that ISR1 “has the highest degree of similarity to the Raf proto-oncogene from mouse”. This statement is demonstrably false. Unfortunately, the only other paper on ISR1 (Mehlgarten 2007) repeats this claim, adding credibility to it. This manuscript should break this chain of errors and not repeat it anymore. ISR1 is NOT related to Raf. Mammalian Raf kinases are members of the tyrosine kinase-like subfamily (Manning 2002). Yeast does not contain any members of this family. ISR1 is not the orthologue of Raf. Of the 117 kinases in yeast, 79 show greater homology to the kinase domain of mouse Raf than ISR1. The 30% identity reported in the Miyahara paper is extremely misleading since that homology is limited to a 109 residue peptide and not the entire kinase domain. The misinformation about a Raf-ISR1 connection needs to stop now. Further, the introduction should delete the whole paragraph on Raf kinase and Ras signaling. A more accurate and factually correct description of ISR1 and its place in the kinomes of yeast and humans would be greatly appreciated. ISR1 is an unusual kinase and has variant sequences in the place of what are usually invariant kinase signature motifs. For instance, the catalytic aspartate that these authors mutate (D280) is usually in the “HRD” motif. In ISR1, this sequence is HGD. Because of its unusual sequence, many databases list ISR1 as a “putative protein kinase”. That is much more accurate and pertinent to this paper than any discussion of Raf.

The connection of ISR1 to Raf and Ras signaling reappears in the discussion where the authors note that the closest relative of ISR1 in S. pombe is the kinase byr2. This is true, but the similarity is very weak and not reciprocal. S. pombe byr2 is more likely to be the orthologue of S. cerevisiae Ste11. If you BLAST pombe byr2 against the S. cerevisiae proteome, the top hit is STE11, not ISR1. In fact, 103 of the 117 yeast kinases show stronger homology to pombe byr2 than does ISR1. S. pombe does not have a clear orthologue of ISR1; there is no connection to Raf or RAS. This paragraph in the discussion should be removed.

2. Staurosporine. ISR1 will be forever linked to staurosporine since ISR stands for “Inhibition of Staurosporine Resistance”. Miyahara et al. in 1998 describe staurosporine as “a potent inhibitor of protein kinase C”. This may be true but it is not the whole story. We now know that staurosporine is a potent inhibitor of almost all protein kinases. It binds to the ATP binding pocket with higher affinity than ATP. In fact, its presence in the ATP pocket stabilizes kinase domain structures explaining why this compound is included during crystallization of so many protein kinases. Probably the most relevant data in the Miyahara paper is that isr1 mutants exacerbate a pkc1 mutant, thus connecting both kinases to the cell wall integrity pathway. Perhaps when referring to staurosporine as a PKC inhibitor, they could add that this compound is really a broad spectrum kinase inhibitor.

3. PSTAIR. Many western blot experiments in this paper use PSTAIR antibody as a loading control (Figs 1,5 and 6). This needs to be explained. My understanding is that PSTAIR is a monoclonal that recognizes the PSTAIR sequence in cdk kinases. Yeast has 2 cdk’s with the PSTAIR sequence, Cdc28 and Pho85. These two proteins have similar molecular weights: 34.0 vs 34.9 kDa. What is being recognized in the PSTAIR blots? What is the MW range being shown in the PSTAIR panels? This may be an odd control blot to choose since Pho85 becomes an important player in the control of ISR1 abundance. There is no mention of the PSTAIR panel in the figure legends.

4. ISR1 deletion details. The authors note that the isr1∆:KAN allele in the MATa collection has RNA processing phenotypes caused by interference with the adjacent gene. They make a new isr1∆ strain with a different marker. It would be worth including the details of how their new deletion allele differs from the KAN allele. How did they make an isr1 deletion that doesn’t interfere with the adjacent gene?

5. Improving the discussion. I have noted above some topics that should be removed from this paper. The one topic that should be added is a better discussion of how their new data relates to the data in the Mehlgarten paper. That paper is the best description of ISR1 to date and connects ISR1 to the cell wall and PKC1. It’s a bit unfair to the Mehlgarten authors to call ISR1 a “previously uncharacterized kinase” (line 418). Poorly characterized, yes; uncharacterized, no.

Reviewer #2: Review PloS Genetics

Although protein glycosylation is an essential and well-studied cellular process, how protein glycosylation might be regulated throughout the cell cycle and in response to stress is not clear. Here, this report focuses on a protein kinase and negative regulator of an early step in protein glycosylation, which is the biosynthesis of an N-linked sugar precursor. This negative regulatory kinase is regulated by the alternate CDK Pho85, which allows cells to alter the levels of the regulator according to the cell’s nutrient needs.

There are many strengths to this study. It is very well written. The data is clear and carefully interpreted. The findings are impactful and presented in a clear and logical manner. The combination of phenotypic and genetic data with immunoblot and MASS SPEC data, plus previous data from the lab and the community lends credence to the study.

The main piece of data that is missing is direct demonstration that Isr1 phosphorylates Gfa1. Typically, this can be shown in two ways, a shift in protein levels by immunoblot (e.g. phos-tag) analysis (which itself is indirect) and by in vitro kinase assays. The authors do present MASS SPEC data to address this point.

Below are some minor comments.

1. Are there homologs to Isr1?

2. Immunoblots can be quantitated. By adjusting levels to the loading control, setting it to 1 and comparing other levels to that (Fig. 1B). Numbers can be represented below the IBs.

3. What is the significance of the fact that the levels of overexpressed ISR1 are the same as Pkc1 in the cell?

4. Section heading and within the paragraph “Isr1 overexpression is synthetic lethal with enzymes in the hexosamine biosynthesis pathway’ should be synthetically

5. Does the fact that Gfa1 -3A cells are sensitive to tunacamycin imply that phosphorylation of Gfa1 is important for balancing the cell’s protein glycosylation needs?

6. Given that Isr1 is cell cycle regulated, it might be phosphorylated by the cell’s major CDK, Cdc28?

7. Plate labels should appear above not below the pictures.

8. In Figure 5, it looks like cdc4-1 is much more important than cdc53-1 in stablilzation of Isr1 levels. Again, quantitation of protein levels might help this analysis.

9. wildtype should be wild-type

10. What do the authors think about all the other proteins that are being phosphorylated by Isr1?

11. Are the glyosylation profiles of select glycosylated proteins impacted by these mutants, observed for example by IB analysis?

Reviewer #3: This is a well-written and compelling story that reveals a function for the yeast Raf homolog, Isr1. Deletion of ISR1 confers tolerance to N-glycosylation inhibitor tunicamycin, whereas its moderate overexpression confers tolerance to the chitin antagonist calcofluor white. Based on these phenotypes, the authors pursued the possibility that ISR1 is a negative regulator of hexosamine biosynthesis, specifically UDP-N-acetyl glucosamine (UDP-GlcNAc) production by GFA1. UDP-GlcNAc is required for the first steps in N-glycosylation and GPI-anchor biosynthesis, as well as for the production of chitin. The authors showed that strong ISR1 overexpression causes growth inhibition, which was suppressed either by over-expression of GFA1, or by addition of glucosamine to the medium, which is scavenged to produce UDP-GlcNAc independently of GFA1.

The authors go on to show that Gfa1 is phosphorylated on neighboring residues in an Isr1-dependent manner, mutation of which to Ala (GFA1-3A) overcame the growth inhibition associated with overexpression of ISR1, supporting the conclusion that Isr1 phosphorylation of Gfa1 is inhibitory. One result that was not entirely consistent with this conclusion was the finding that the GFA1-3A mutation did not confer tolerance to tunicamycin (Fig. 4E), as would be expected if it was to phenocopy deletion of ISR1. The authors explain this with a demonstration that the GFA1-3A mutation is somewhat hypomorphic and causes tunicamycin sensitivity by itself. They further suggest that Isr1 has other targets that contribute to its role in N-glycosylation. This is a reasonable conjecture.

Finally, the authors show that the N-terminal regulatory domain of Isr1 contains a phosphodegron that is phosphorylated by Pho85 and targeted by the SCF-Cdc4 ubiquitylation complex. This confers tight cell cycle regulation of Isr1 levels so that its activity is largely confined to the G1/S transition. These experiments set out in Fig. 5, should be accompanied by quantitation of the immunoblot bands. Aside from this lone criticism, this manuscript is viewed to be very strong.

**Have all data underlying the figures and results presented in the manuscript been provided?**

Reviewer #1: Yes

Reviewer #2: Yes

Reviewer #3: Yes

PLOS authors have the option to publish the peer review history of their article (what does this mean?). If published, this will include your full peer review and any attached files.

Reviewer #1: No

Reviewer #2: No

Reviewer #3: No

---

## [Editor Report · Decision Letter 1]

8 May 2020

Dear Dr Toczyski,

We are pleased to inform you that your manuscript entitled "The kinase Isr1 negatively regulates hexosamine biosynthesis in S. cerevisiae" has been editorially accepted for publication in PLOS Genetics. Congratulations!

Yours sincerely,

Jason MacGurn

Guest Editor

PLOS Genetics

Gregory P. Copenhaver

Editor-in-Chief

PLOS Genetics

Comments from the reviewers (if applicable):

**Data Deposition**

http://datadryad.org/submit?journalID=pgenetics&manu=PGENETICS-D-20-00243R1

**Press Queries**

---

## [Editor Report · Acceptance letter]

18 Jun 2020

PGENETICS-D-20-00243R1 

The kinase Isr1 negatively regulates hexosamine biosynthesis in <I>S. cerevisiae</I> 

Dear Dr Toczyski, 

We are pleased to inform you that your manuscript entitled "The kinase Isr1 negatively regulates hexosamine biosynthesis in <I>S. cerevisiae</I>" has been formally accepted for publication in PLOS Genetics! Your manuscript is now with our production department and you will be notified of the publication date in due course.

With kind regards,

Kaitlin Butler

PLOS Genetics

On behalf of:
